# A Regulatory Noncoding RNA, nc886, Suppresses Esophageal Cancer by Inhibiting the AKT Pathway and Cell Cycle Progression

**DOI:** 10.3390/cells9040801

**Published:** 2020-03-26

**Authors:** Wonkyun Ronny Im, Hyun-Sung Lee, Yeon-Su Lee, Ju-Seog Lee, Hee-Jin Jang, Seon-Young Kim, Jong-Lyul Park, Yeontaek Lee, Moon Soo Kim, Jong Mog Lee, In-Hoo Kim, Sung Ho Jeon, Yong Sun Lee

**Affiliations:** 1Department of Cancer Biomedical Science, Graduate School of Cancer Science and Policy, National Cancer Center, Goyang 10408, Korea; 74920@ncc.re.kr (W.R.I.); ikim@ncc.re.kr (I.-H.K.); 2Department of Life Science and Multidisciplinary Genome Institute, Hallym University, Chuncheon 24252, Korea; dusxor123@gmail.com; 3Division of Thoracic Surgery, Michael E. DeBakey Department of Surgery, Baylor College of Medicine, Houston, TX 77030, USA; Hyun-Sung.Lee@bcm.edu (H.-S.L.); heejinj@bcm.edu (H.-J.J.); 4Rare Cancer Branch, Research Institute, National Cancer Center, Goyang 10408, Korea; yslee2@ncc.re.kr; 5Department of Systems Biology, University of Texas MD Anderson Cancer Center, Houston, TX 77030, USA; jlee@mdanderson.org; 6Medical Genomics Research Center, KRIBB, Daejeon 34141, Korea; kimsy@kribb.re.kr (S.-Y.K.); nlcguard@kribb.re.kr (J.-L.P.); 7Center for Lung Cancer, National Cancer Center, Goyang 10408, Korea; vsd10@ncc.re.kr (M.S.K.); jongmog@ncc.re.kr (J.M.L.)

**Keywords:** nc886, esophageal cancer, cell cycle, AKT, prognosis

## Abstract

nc886 is a regulatory non-coding RNA (ncRNA) whose expression is frequently silenced in malignancies. In the case of esophageal squamous cell carcinoma (ESCC), nc886 silencing is associated with shorter survival of patients, suggesting nc886’s tumor suppressor role in ESCC. However, this observation has not been complemented by an in-detail study about nc886’s impact on gene expression and cellular phenotypes. Here we have shown that nc886 inhibits AKT, a key protein in a renowned pro-survival pathway in cancer. nc886-silenced cells (nc886^−^ cells) have activated AKT and altered expression of cell cycle genes. nc886^−^ cells tend to have lower expression of CDKN2A and CDKN2C, both of which are inhibitors for cyclin-dependent kinase (CDK), and higher expression of CDK4 than nc886-expressing cells. As a result, nc886^−^ cells are hyperactive in the progression of the G1 to S cell cycle phase, proliferate faster, and are more sensitive to palbociclib, which is a cancer therapeutic drug that targets CDK4/6. Experimentally by nc886 expression and knockdown, we have determined the AKT target genes and cell cycle genes that are controlled by nc886 (nc886-associated gene sets). These gene sets, in combination with pathologic staging and nc886 expression levels, are a vastly superior predictor for the survival of 108 ESCC patients. In summary, our study has elucidated in ESCC how nc886 inhibits cell proliferation to explain its tumor suppressor role and identified gene sets that are of future clinical utility, by predicting patient survival and responsiveness to a therapeutic drug.

## 1. Introduction

nc886 is a medium-sized (101 nucleotides long) non-coding RNA (ncRNA) that controls gene expression by binding to target proteins and modulating their activities [1]. This gene-regulatory mechanism is distinct from microRNAs (miRNAs) or nuclear long ncRNAs that bind to complementary target RNA or DNA (reviewed in [2]). nc886 has begun to gain attention since its expression is dysregulated in cancer. nc886 is transcribed by RNA polymerase III (Pol III) that also transcribes medium-sized ncRNAs with indispensable cellular functions, including 5S ribosomal RNA, transfer RNAs, and U6 type small nuclear RNA. As a Pol III transcript, the expression of nc886 is ubiquitous and low in normal human tissues but is generally elevated during tumorigenesis [1,3]. One interesting feature of nc886 expression is its frequent silencing in malignancies by CpG DNA methylation at the promoter region.

Esophageal cancer (EC) is one of the deadly cancers. Worldwide, EC ranks eighth in the incidence rate and sixth in disease-associated death [4]. Esophageal squamous cell carcinoma (ESCC) is the major histological subtype of EC, especially in East Asian countries. A therapeutic strategy to combine surgery, chemotherapy, and radiotherapy has improved the survival of ESCC patients. Nonetheless, the five-year overall survival (OS) is no higher than 20%, mostly because it is challenging to identify ESCC patients at a high risk of recurrence. This poor prognosis calls attention to the need for reliable molecular markers. Considerable efforts have been made to identify prognostic biomarkers, which include cyclooxygenase-2, vascular endothelial growth factor, survivin, and inhibitors for cyclin-dependent kinases (CDKs) (reviewed in [5]). We have recently identified nc886 to be a promising prognostic marker in ESCC, by showing that nc886 silencing in ESCC patients is associated with their short survival [6,7].

These patient survival data suggest that nc886 is a tumor suppressor in ESCC. However, nc886’s role in ESCC has not yet been examined in sufficient detail. nc886’s impact on cancer cannot be simply defined, because nc886 expression is of two opposite directions: increasing vs. silencing, as stated above. Not only its expression, but also its function appears to be two-faced depending on cancer types and stages. We and others have found that patients with high nc886 survive longer in the case of gastric cancer and acute myeloid leukemia as well as ESCC [6,8,9], whereas ovarian cancer patients with high nc886 have a shorter survival time [10]. According to our current knowledge, nc886 inhibits the activity of two proteins, protein kinase R (PKR) and dicer, both of which are implicated in tumorigenesis. Although nc886’s oncogenic role is explained by its inhibition of dicer in the case of ovarian cancer, nc886’s tumor suppressor role in ESCC could not be explained by either of them (to be elaborated in Section 3) and so the underlying mechanism has remained elusive. Herein, we have investigated the effect of nc886 on cell proliferation to provide a clue as to how nc886 silencing promotes ESCC carcinogenesis. Furthermore, we have tested a prognostic role of novel gene signatures related to nc886’s function to control the activity of AKT and the expression of cell cycle genes in a cohort of ESCC patients.

## 2. Materials and Methods

### 2.1. Cell Lines, Plasmid DNAs, Antibodies, and Other Reagents

The HEK-293T cell line was purchased from American Type Culture Collection (ATCC, Manassas, VA, USA) An immortalized esophageal epithelial cell line (Het-1A) and three ESCC cell lines (TE-1, TE-8, and TT) were used in this study. All cell lines were provided by Drs. Xiaochun Xu and Julie J. Izzo at the University of Texas MD Anderson Cancer Center. Cell lines were validated by short tandem repeat (STR) DNA fingerprinting using the AmpF_STR Identifiler kit (Applied Biosystems, Grand Island, NY, USA) according to manufacturer’s instructions. The STR profiles were compared to known ATCC fingerprints (http://www.atcc.org/) and the Cell Line Integrated Molecular Authentication database (CLIMA, version 0.1.200808, http://bioinformatics.istge.it/clima/). The STR profiles matched known DNA fingerprints or were unique. Cell lines were tested for mycoplasma contamination and confirmed to be negative.

The plasmid “pLPCX-U6” was constructed by inserting the U6 promoter into the pLPCX vector (Clontech Laboratories Inc., Mountain View, CA, USA) and “pCAGGS-GFP” was constructed by subcloning of the green fluorescent protein (GFP) gene into the pCAGGS vector. We inserted nc886 DNA fragments (101-nt long and 649-nt long, as depicted in Figure 1A) into these control vectors (“pLPCX-U6” and “pCAGGS-GFP”), to generate nc886-expressing plasmids (“pLPCX-U6:nc886” and “pCAGGS-GFP/nc886” respectively). We transfected these plasmids into HEK-293T cells by Lipofectamine™ 2000 reagent (Invitrogen, Carlsbad, CA, USA), isolated clones by a standard laboratory protocol, and established nc886^+^ stable lines or nc886^−^ control cell lines (for their names, see Figure 1A). The detailed information about plasmids and the protocol is available upon request.

Antibodies against phospho-AKT at Ser473 (cat # 9271), total AKT (cat # 4691), phospho-FOXO3 at Thr32 (cat # 9464), total FOXO3 (cat # 2497), and GAPDH (cat # 2118) were purchased from Cell Signaling Technology (CST, Beverly, MA).

### 2.2. ESCC Patients

We enrolled 108 patients who: (1) had histologically confirmed to be ESCC confined to the thorax, (2) had undergone complete esophagectomy with adequate lymph node dissection and did not receive any perioperative chemotherapy or radiotherapy, (3) had available paired tumor and adjacent normal tissue in the tumor bank, and (4) were followed-up completely. We excluded patients who: (1) received perioperative chemotherapy and/or radiotherapy at any period except confirmation of recurrence, (2) had a prior cancer diagnosis, (3) had double primary synchronous cancer, (4) had cervical esophageal cancer, or (5) were not monitored completely. Regular follow-up was conducted via telephone or mail twice a year, in April and October. A chest computed tomography (CT) scan was routinely performed during the first follow-up visit after discharge. In addition, all patients underwent regular evaluations including a routine blood examination, chest x-ray, and chest CT every 3 months during the first two years. Subsequently, all patients were monitored annually. Positron emission tomography-CT (PET-CT) and esophagogastroduodenoscopy were performed annually or more frequently if necessitated according to clinical history and clinical examination findings. All human samples were collected with informed patient consent, and the study protocol was approved by the institution’s ethics committee. After obtaining fresh frozen tissues, total RNA was extracted using the mirVana^TM^ miRNA Isolation Kit (Ambion Inc., Austin, TX, USA) according to the manufacturer’s procedures.

### 2.3. RNA Measurement

Total RNA from cell lines was isolated by Trizol reagent (Life Technologies, Carlsbad, CA, USA). cDNA was synthesized by the amfiRivert kit (GenDEPOT, Barker, TX, USA) and was amplified by AccuPower^®^ Taq PCR PreMix (BIONEER, Daejeon, Korea). RNA expression levels were assessed by visualizing PCR products on an agarose gel or by real-time qRT-PCR with AccuPower^®^ GreenStar™ qPCR MasterMix (BIONEER) and Exicycler^TM^ 96 Real-Time Quantitative Thermal Block (BIONEER). Primer sequences are (5′ to 3′): “ggccaaggtcatccatgacaactt” and “tagaggcagggatgatgttctgga” for GAPDH, “cgggtcggagttagctcaagcgg” and “aagggtcagtaagcacccgcg” for nc886, “ggcagtaaccatgcccgcatagat” and “catgcctgcttctacaaacccaca” for CDKN2A, “aatggatttggaaggactgcgctg” and “aaagtgtccaggaaacctgctctg” for CDKN2C, “actcagctgtcctccaggttcaaa” and “tgtccttttcaccatccactcgct” for CUL1, and “cctatcactcagtcggtgctatga” and “aacagttgaagggtaccatctggc” for SKP2.

### 2.4. nc886 Knockdown (KD)

Anti-oligos (“anti-nc886” and “anti-control”) used in this study were the same as our previous study [6] (therein named “anti886 75-56” and “anti-vt 21-2” respectively). A total of 100 nM of each anti-oligo was transfected using Lipofectamine™ 3000 reagent (Invitrogen, Carlsbad, CA) per the manufacturer’s instruction.

### 2.5. mRNA Expression Profiling (Array)

Total RNA from 293T-U6 and 293T-U6:nc886 cells (designated as “nc886-EXP”) was subjected to Affymetrix Human Genome U133 Array (Affymetrix, Santa Clara, CA, USA). For nc886 KD in Het-1A cells and ESCC patients, array data had been generated in our previous studies [6,7] and deposited to the Gene Expression Omnibus database (GSE66258 for ESCC samples and GSE51732 for Het-1A) in the National Center for Biotechnology Information. We re-analyzed these data in this study. It should be mentioned that the array platform had been HumanHT-12 v4 Expression Beadchip Kit (Illumina, San Diego, CA, USA), which was discontinued in 2016. Therefore, we inevitably employed the Affymetrix platform for nc886-EXP, albeit different from the Illumina platform for nc886 KD and patients.

### 2.6. Reverse Phase Protein Arrays (RPPA)

RPPA data were generated in the Department of Systems Biology at MD Anderson Cancer Center. Briefly, protein was extracted using RPPA lysis buffer (1% Triton X-100, 50 mM HEPES (pH 7.4), 150 mM NaCl, 1.5 mM MgCl_2_, 1 mM EGTA, 100 mM NaF, 10 mM Na-pyrophosphate, 1 mM Na_3_VO_4_, 10% glycerol, 1 mM phenylmethylsulfonyl fluoride, and 10 μg/mL aprotinin) as described previously [11]. Cell lysates were initially adjusted to 1 μg/μL, diluted serially in 5-fold in the lysis buffer, and printed on nitrocellulose-coated slides (Grace Bio-Labs, Bend, OR, USA). Slides were probed with 172 validated primary antibodies followed by corresponding secondary antibodies (goat anti-rabbit IgG, goat anti-mouse IgG, or rabbit anti-goat IgG). Signal was detected using diaminobenzidine colorimetric reaction and normalized by using the fitted curve (“supercurve”) approach.

### 2.7. Measurement of Cell Proliferation, Cell Synchronization, and Cell Cycle Analysis

Cell numbers were counted directly on a hemocytometer under a microscope or by flow cytometry (FACSCanto™ II; BD Biosciences, San Jose, CA, USA).

Synchronization of cells at the G2/M phase was done by treatment with 100 ng/mL of nocodazole (Sigma-Aldrich, St. Louis, MO, USA) for 18 h. After replacement with nocodazole-free media, cells were collected at 0, 6, and 12 h for measurement of DNA content. Cells were fixed with 70% ethanol, DNA was stained with propidium iodide, and the DNA content was analyzed by using (FACSCanto™ II). FACS data were analyzed by FACS Diva 7 software (BD Bioscience, San Jose, CA, USA).

### 2.8. Cell Viability and Apoptosis Assay upon Drug Treatment

Palbociclib (PD0332991) HCl (catalog no. S1116) was purchased from Selleckchem (www.selleckchem.com). The MTT (3-(4,5-dimethylthiazol-2-yl)-2,5-diphenyltetrazolium bromide) assay (CellTiter 96^®^ Non-Radioactive Cell Proliferation Assay; Promega, Madison, WI, USA) was utilized to assess the effect of palbociclib on cell viability. Cells were seeded in culture media (3 × 10^4^ cells/well) in triplicate in a 96-well plate and incubated overnight. Palbociclib was added at eleven different concentrations (0, 0.15625, 0.3125, 0.625, 1.25, 2.5, 5, 10, 20, 40, and 80 µM) at 37 °C. After 48 h incubation, the cells were washed with fresh media and were grown in palbociclib-free media for an additional 24 h, until MTT assays were done as specified by the manufacturer’s instructions. IC_50_, a concentration at which cell viability was down to half, was calculated using Prism 5.0 (GraphPad Software Inc., San Diego, CA, USA).

For apoptosis assays, 10^6^ cells were initially plated on 6-well plates and incubated overnight. Cells were treated with 5 µM of palbociclib, incubated for 24 h, and subjected to apoptosis assay with flow cytometry by using Annexin V-FITC and 7-AAD (catalog no. 640922; BioLegend, San Diego, CA, USA). All experiments were independently repeated in quadruplicate.

### 2.9. Statistical Analysis

Unless otherwise specified, routine biological assays such as cell counting, MTT, and qRT-PCR (for Figures 1C,F, 4D and 7C,D,F) were conducted in triplicates from which a mean and the standard error were calculated. Student’s T-test was employed to calculate a *p*-value. Enrichment *p*-values were calculated by Fisher’s exact test.

Gene set enrichment analysis (GSEA: http://software.broadinstitute.org/gsea/index.jsp) was performed to determine whether a priori defined set of genes shows statistical concordance according to nc886 expression [12]. We analyzed correlations between nc886 expression and microarray data with Pearson’s correlation coefficient. Survival curves were generated with the Kaplan–Meier’s method and intergroup comparisons were performed with the log-rank test. Disease-specific survival (DSS) was defined as the time from surgery to death related to esophageal cancer, recurrence-free survival (RFS) as the time from surgery to the first confirmed recurrence as well as death, and overall survival (OS) as the time from surgery to death. Univariable Cox regression analysis was used to determine whether variables were associated with survival. We used nc886 expression, gene signatures, and tumor-node-metastasis (TNM) stage as variables. C-statistics in R language and software environment (http://www.r-project.org) was used to calculate the concordance index between variables and survival. Statistical significance was accepted when *p* < 0.05, and all tests were two-tailed. All statistical analyses were performed with SPSS 25.0 (released 2017. IBM SPSS Statistics for Windows, Version 25.0; IBM Corp., Armonk, NY, USA).

## 3. Results

### 3.1. nc886 Inhibits Cell Proliferation

As stated in the Introduction, our previous patient data indicate that nc886 is a putative tumor suppressor in ESCC. To study the mechanistic detail, loss-of-function, and gain-of-function phenotypes need to be assessed in esophageal cell lines. We performed nc886 knockdown (KD) in Het-1A, a non-malignant esophageal cell line that expresses nc886 (designated as nc886^+^ cells), expecting a more tumorigenic phenotype (such as increased cell growth) [6]. Conforming to this expectation, nc886-KD provokes several oncogenes. However, it also leads to the activation of PKR and resultant apoptosis, in line with nc886’s well-studied role as an inhibitor of PKR that is a pro-apoptotic protein. The PKR-mediated apoptosis eclipses all other effects of nc886-KD on Het-1A cells and makes any further experiments impractical. Then, we switched to the gain-of-function approach. nc886 expression has become low or epigenetically silenced in ESCC cells (nc886^−^ cells) [6] and we attempted to construct an isogenic nc886^+^ ESCC cell line from them. Nonetheless, we could not isolate any nc886^+^ clone, because of nc886’s anti-proliferative effect on ESCC cells. When we forced nc886 expression in two ESCC cell lines, TE-1 and TE-8, by transient transfection of nc886-expressing DNA, cell proliferation was impaired as early at 24 h (Appendix A). These data indicated that these ESCC cells were addicted to the nc886^−^ status and could not proliferate when artificially made to be nc886^+^. Inevitably, we looked into a surrogate and decided to use HEK-293T (shortly “293T”), a human embryonic kidney cell line transformed by SV40 T antigen [13]. The cell line 293T was chosen as a last resort but appeared to be a legitimate alternative because nc886’s impact on gene expression was similar between 293T and Het-1A cells (to be shown later).

We constructed two different versions of nc886^+^ 293T cell lines and also corresponding vector control lines (see Figure 1A for their nomenclature) and confirmed nc886 expression by RT-PCR measurement (Figure 1B). While culturing these cells, we sensed that 293T-U6:nc886 and 293T-GFP/nc886 cells grew slowly as compared to 293T-U6 and 293T-GFP cells respectively. Since active cell proliferation is a hallmark event during the transformation process, we focused on this phenotype in this study. The number of 293T-U6 cells was ~1.5-fold more than 293T-U6:nc886 cells at 4 days after the same number of cells were initially plated (Figure 1C). We also conducted a cell-mixing experiment by taking advantage of GFP expression in 293T-GFP/nc886 cells. In this experiment, GFP-expressing (GFP^+^) cells (either 293T-GFP/nc886 or 293T-GFP) were mixed with the equal number of the original 293T cells which were GFP-negative (GFP^−^), followed by monitoring the ratio of GFP^+^/ GFP^−^ (Figure 1D for the experimental scheme). GFP^+^ cells were depleted as the co-culture continued, and importantly, this depletion was more severe in 293T-GFP/nc886 than in 293T-GFP (Figure 1E,F). The smaller number of nc886^+^ cells than nc886^−^ cells might have resulted from increased cell death. However, the portion of apoptotic 293T-U6:nc886 cells was negligible in our Annexin V staining (Appendix A). Collectively these data clearly showed that nc886^+^ cells proliferate slowly.

### 3.2. nc886^+^ Cells Have a Longer G1 Duration than nc886^−^ Cells

We hypothesized that the retarded growth of nc886^+^ cells was due to a delay in a particular phase in the cell cycle [14]. To test this idea, we examined cell cycle profiles by flow cytometry, after treating asynchronously growing cells with nocodazole, a microtubule inhibitor (Figure 2A). Upon nocodazole treatment, cells were synchronized at the G2/M phase, as confirmed by a single peak at the DNA content of 2n. At 6 h after removal of nocodazole, another peak at the 1n position appeared, indicating that cells were released from G2/M and entered into the G1 phase. nc886^+^ and nc886^−^ cells displayed a nearly identical pattern at this time point. At 12 h, nc886^−^ cells progressed further to the S phase and this progression was delayed in the case of nc886^+^ cells (the 293T-U6 and 293T-U6:nc886 pair in Figure 2; the 293T-GFP and 293T-GFP/nc886 pair in Appendix A). Our data indicated that nc886 procrastinated the G1-to-S transition but did not affect the G2/M-to-G1 transition. The extended G1 duration of nc886^+^ cells explains why they proliferated more slowly than nc886^−^ cells.

### 3.3. nc886 Controls Cell Cycle Genes, which Is Probably the Reason for the Extended G1 Period of nc886^+^ Cells

Since nc886 has a gene-regulatory function, we examined a gene expression pattern to obtain a clue as to why nc886^+^ cells had a longer G1 than nc886^−^ cells. We ran a microarray in the pair of cell lines used in Figure 1 and Figure 2 (termed “nc886-EXP”, 293T-U6:nc886 values relative to 293T-U6). It should be noted that a considerable fraction of altered genes in nc886-EXP would be secondary consequences because these cells were maintained for several months until using the microarray experiment. Therefore, we complemented nc886-EXP data with nc886 KD data (termed “nc886-KD”, anti-nc886 values relative to anti-control). nc886-KD had been done in Het-1A cells by transiently transfecting anti-nc886 (an oligonucleotide that targets and suppresses nc886) for just a couple of days [6]. Therefore, nc886-KD data will represent nc886’s direct consequences better than nc886-EXP.

Having array data, we attempted to compare nc886-EXP and nc886-KD. However, different array platforms between them (Figure 3A) imposed an impediment in direct gene-to-gene comparison. We also wanted to know which biological pathways or transcription factor (TF) activities were altered, to obtain a clue as to how nc868-EXP leads to delayed G-S transition. To this aim and also for comparison between nc886-EXP and -KD, we analyzed array data against the Molecular Signature Database (MSigDB: http://software.broadinstitute.org/gsea/msigdb) (Figure 3A).

MSigDB has several sets of gene collections based on sequence motifs, pathways, etc. For example, TFT is a collection of target genes for TFs. There are 615 TFT sets, each of which contains several tens to hundreds of genes that harbor a binding site(s) for the corresponding TF in their promoter regions. Another example is 674 sets of genes (674 Reactome sets) defined by the Reactome pathway database. Overall enrichment or depletion of a given set is calculated from array values of genes in the set and is expressed as a positive or negative Z-score. For Reactome gene sets, we obtained 674 Z-scores from nc886-EXP and nc886-KD, compared them in a scatter plot, and observed a significant negative correlation (Figure 3B and Appendix A). Likewise, nc886-EXP and nc886-KD TFT Z-scores were also negatively correlated (Figure 3C and Appendix A). These results suggested that nc886 imposed an impact on the gene expression and cellular pathways of 293T cells similarly on those of the esophageal cell line Het-1A. This notion was also supported by reciprocal comparison of altered genes. When we compared the most increased and decreased 1000 genes from nc886-EXP and nc886-KD respectively (or vice versa), they overlapped at a statistically significant level (Appendix A). Collectively, all these data relieved our initial apprehension of whether 293T was suitable to investigate nc886’s tumor suppressor role in ESCC.

We looked over individual gene sets to see which pathway was most affected by nc886. In Reactome sets, nine pathways were significantly (with a Z-score cutoff > 3) activated (blue bars in Figure 3D; see also Appendix A) in nc886-EXP. Of note, most of them (8 out of 9) were somehow related to cell proliferation and three were directly relevant to the G1/S phase (Figure 3D, highlighted in red brackets). Concurring with their up-regulation in nc886-EXP, all the eight pathways had significant negative Z-scores (< −3) in nc886-KD.

Next, we wanted to investigate individual cell cycle genes that were controlled by nc886. For this, we retrieved 40 genes from the G1 phase subset in the Reactome set, intersected them with 6563 genes that were significantly (*p* < 0.05) changed both in nc886-KD and nc886-EXP, and obtained 19 genes (“nc886/G1-associated genes”, Figure 4A). These genes were analyzed in a cohort of 108 ESCC patients for which nc886 qRT-PCR values and mRNA array data were available [7]. We found a majority (14 out of 19) of nc886/G1-associated genes obtained from cell line data to be significantly (*p* < 0.01) associated with nc886 also in the patient data (Figure 4B).

Upon scrutinizing the 14 genes, we chose CDKN2A (also known as p16INK4a or p14ARF), CDKN2C (also known as P18-INK4C or P18-INK6), SKP2, and CUL1 for further assessment. CDKN2A and CDKN2C restrict cells at G1 by inhibiting CDK4/CDK6, whereas SKP2 and CUL1 promote the G1-S transition by degrading CDK inhibitors [15]. In line with our earlier finding that nc886^−^ cells are more proficient at G1-S transition than nc886^+^ cells, the expression of CDKN2A and CDKN2C was positively correlated with nc886 in the 108 ESCC patients (Figure 4C) and was also upregulated in 293T-U6:nc886 cells (Figure 4D). In the case of SKP2 and CUL1, they were negatively associated with nc886 (Figure 4C,D).

Collectively from our data, nc886 delays G1-S transition by controlling cell cycle genes. Since nc886 has been shown to inhibit PKR and dicer [1,10], we interrogated whether the delayed G1 in nc886-EXP could be explained by these two proteins. PKR is a protein implicated in cell proliferation and its representative downstream event is the activation of nuclear factor- κB (NF-κB) (reviewed in [16]). Therefore we examined NF-κB in the MSigDB TFT sets. Consistent with our previous studies, nc886-KD activated PKR and consequently NF-κB, as indicated by positive Z-scores (Appendix A). However, NF-κB was not suppressed in nc886-EXP. As described earlier, PKR induces apoptosis when activated and thus should stay in a latent state in naturally proliferating cells [16]. In this situation, ectopic expression of nc886 and thereby further inhibition of PKR would barely have an effect. In summary, although activated by nc886-KD, PKR could not explain the cell cycle phenotype of nc886-EXP. We also investigated dicer, a key enzyme for miRNA biogenesis, by looking into MSigDB MIR sets, a collection of miRNA target genes. We could not see an inhibitory effect of nc886 on the miRNA pathway in nc886-EXP (Appendix A).

### 3.4. nc886 Suppresses the AKT Pathway

Neither PKR nor dicer could account for nc886’s role in the cell cycle progression. Therefore, we attempted to identify another protein target of nc886. Toward this goal, we measured a proteome repertoire by employing a high-throughput platform called RPPA. In this experiment, cell lysates were placed for hybridization on a nitrocellulose-coated slide that has 172 antibodies are spotted, to comprehensively measure corresponding proteins. These antibodies include proteins and their phospho-forms in well-known pathways. We ran RPPA upon nc886-KD, because short-term KD is more advantageous than nc886-EXP when discerning nc886’s direct effect, as aforementioned. We obtained expression values of 172 proteins and found 23 to be changed at a statistically significant level (*p* < 0.05 from triplicate experiments; Figure 5A and Appendix A).

When we looked through these 23 proteins, we recognized the alteration of proteins in the AKT pathway. AKT proteins are comprised of highly homologous three isoforms (AKT1, AKT2, and AKT3) and their phosphorylation represents the active form [17]. In our RPPA data, phosphorylation of AKT1 at Thr308 and Ser473 was increased, with the total AKT1 being relatively unchanged (Figure 5A). This result was confirmed by Western blot (Figure 5B). In accordance with AKT1 activation, nc886-KD resulted in increased phosphorylation of AKT1 substrate 1 (AKT1S1; also known as PRAS40) and ribosomal protein S6 (RPS6). AKT1S1 is directly phosphorylated by AKT [18]. RPS6 is phosphorylated by ribosomal protein S6 kinase B1 (RPS6KB1; also known as S6K), whose activity is promoted by AKT [19,20]. We obtained these data from nc886-KD, whereas we observed the cell cycle phenotype in nc886-EXP (slow cell proliferation due to a delay at G1; Figure 1, Figure 2). Therefore, we wanted to validate the RPPA data in nc886-EXP. Our Western blot showed that AKT1 in 293T-U6:nc886 cells (designated as nc886^+^ cells in Figure 5C) was less phosphorylated at Ser473 than 293T-U6 cells at 6 and 12 h after release from nocodazole arrest.

In addition to AKT, we measured FOXO3, an AKT target protein. FOXO3 activates transcription of CDK inhibitors such as CDKN1A (also known as p21) and thereby plays an important role in preventing quiescent cells from entering into active cell cycle (reviewed in [21]). When phosphorylated by AKT, FOXO3 is exported to the cytoplasm and consequently loses its activity as a transcriptional activator. Compared to nc886^−^ cells, nc886^+^ cells had lower phospho-FOXO (therefore elevated FOXO activity; see Figure 5C) in line with lower phospho-AKT. This result was in agreement with the positive correlation between nc886 expression and CDKN1A in ESCC patient samples (Figure 4B). Collectively, all of these proved that nc886 represses AKT, the key protein in a renowned oncogenic pathway that promotes the G1-S transition (reviewed in [17,22]), which could explain why nc886^+^ cells had delayed G1-S transition (Figure 2).

AKT activation leads to a change in gene expression. Hence, nc886’s impact on AKT was further evaluated in our gene expression data. We curated a list of 184 AKT downstream genes collected from a recent review article [17] and public databases including Reactome and Biocarta gene sets (http://software.broadinstitute.org/gsea/msigdb/) and RGD (Rat Genome Database; https://rgd.mcw.edu/) (Appendix A). Some of them have been known to be up-regulated and others to be down-regulated when AKT is activated. To apply them to the array data of 108 ESCC patients, we initially selected altered genes from nc886-KD data (Figure 5D). Among the 184 genes, 34 increased genes and 19 decreased genes upon nc886-KD satisfied our criteria (>2-fold changes of gene expression values in Z-score and *p*-values < 0.05; see Appendix A). In GSEA plots, the 34 and 19 genes were negatively and positively correlated to nc886 respectively at statistically significant levels (Figure 5D). The accordance of the nc886/AKT-associated genes between nc886-KD data and patient data suggested that nc886 suppresses the AKT pathway also during in vivo tumorigenesis.

### 3.5. The Association among nc886, AKT, and Cell Cycle Genes is a Survival Predictor for ESCC Patients

Here we found nc886’s role to suppress the AKT pathway and cell cycle progression and identified nc886-associated gene signatures (34 and 19 nc886/AKT-associated genes in Figure 5D; 9 and 5 nc886/G1-associated genes in Figure 4B). We tested the prognostic utility of these genes in the 108 ESCC patients. An AKT signature score was generated by the difference of geometric means between the up-regulated 34 genes and the down-regulated 19 genes upon nc886-KD. A cell cycle signature score was also similarly calculated from the nine genes and the five genes that were negatively and positively correlated with nc886, respectively. Having these scores, the cohort was dichotomized according to each median value. TNM staging and nc886 expression were independent prognostic factors for RFS. The C-statistics showed that the combination of all four factors (nc886, the AKT signature score, the cell cycle signature score, and TNM staging) had the most powerful capacity to predict recurrence and survival (Figure 6A).

Next, we calculated prognostic scores by consecutively summing the TNM stage (stage I and II = 0 vs. stage III and IV = 1), the nc886 expression (high = 0 vs. low = 1 by the median value), the AKT signature score (low = 0 vs. high = 1 by the median value), and the cell cycle score (low = 0 vs. high = 1 by the median value). Stratification by the prognostic score into five subgroups showed distinct DSS, RFS, and OS curves in the cohort of 108 ESCC patients (Figure 6B). In all three curves, a higher prognostic score was proportional to a shorter survival; patients with score 4 (advanced stages III and IV, low nc886, high AKT signature score, and high cell cycle score) survived the worst, whereas those with score 0 (vice versa) survived the best. In summary, the AKT downstream genes and cell cycle genes predicted survival of patients better when combined with nc886 and TNM staging, opening the possibility that these genes could be used as a biomarker for ESCC prognosis.

### 3.6. Therapeutic Implication for ESCC with Unfavorable Prognosis

As a backward approach to proving the association of nc886 with the AKT target genes (Appendix A) and cell cycle genes (Figure 4A), we performed unsupervised clustering of the 108 patients with those genes (Appendix A), to identify two distinct clusters termed “Cluster1” and “Cluster2” (Figure 7A). Cluster2 was featured by enrichment of the majority of genes including CDK4 (Figure 7A) and lower expression of nc886 (Figure 7B). Some cancer drugs target cell cycle genes [23]. For example, palbociclib is an FDA-approved CDK4/6 inhibitor. Therefore, we interrogated whether nc886 expression, which was associated with CDK4, could predict an ESCC cell’s response to palbociclib. When compared to non-malignant Het-1A cells, nc886 expression was moderate in TE-1, low in TE-8, and silenced in TT (Figure 7C). We treated titrating amounts of palbociclib in these three ESCC cell lines and found the order of IC50 values to be TE-1 > TE-8 > TT (Figure 7D). In line with the IC50 data, the apoptosis rate was highest in TT and higher in TE-8, as compare to TE-1 (Figure 7E). Our experimental data were consistent with palbociclib sensitivity information from a public database (Appendix A) and suggested that cells were more sensitive to palbociclib when nc886 expression was low. To prove the causal link between nc886 expression and palbociclib sensitivity, we performed nc886 KD before palbociclib treatment. TE-1 cells transfected with anti-nc886 had a lower IC50 value than anti-control (Figure 7F). All of our data, which were in concordant with our earlier finding that nc886-silenced ESCC cells had a higher CDK activity, provides a potential utility of nc886 as a predictive marker for ESCC patients when palbociclib treatment is considered.

## 4. Discussion

In this study, we have shown that nc886 inhibits cell proliferation by delaying the G1-S transition and that nc886 suppresses the oncogenic AKT pathway and controls cell cycle genes (summarized in Figure 8). To interpret our findings in the context of in vivo ESCC carcinogenesis, several facts about nc886 expression should be recalled. First, normal quiescent cells have a low expression level of nc886 [3]. Second, when neoplastic cells arise and begin to grow at very early stages of carcinogenesis, nc886 expression increases because the Pol III activity is proportional to cell proliferation rates [24]. Third, nc886 silencing is a stochastic event that could occur anytime during carcinogenesis. Based on our data, the early rise of nc886 in neoplastic cells will have a suppressive effect on their proliferation and might contribute to lowering a cancer incident rate as a checkpoint mechanism. When nc886 silencing occurs in a neoplastic cell, it will lead to AKT activation and promotion of the G1-S transition. As a result, this nc886^−^ cell will have a growth advantage and outgrow normal or other neoplastic cells to eventually develop into a clinically detectable tumor. This scenario was supported by our data from a cohort of 108 ESCC patients. nc886 silencing occurred in a significant fraction of the patients and, reasonably due to the growth inhibitory role of nc886, the prognosis of those nc886^−^ patients was poor. Furthermore, we provided a basis for clinical trials for CDK4/6 inhibitors in ESCC patients with low nc886 and poor prognosis.

In our effort to elucidate how nc886 causes the delayed G1-S transition, we analyzed our array data to find that nc886^+^ cells have altered expression of CDK inhibitors and other cell cycle genes (CDKN2A, CDKN2C, CDKN1A, etc.; Figure 4B–D). We also found that nc886 suppressed AKT activity (Figure 5). An outstanding question is the nature of the causal and mechanistic link among nc886, AKT, and cell cycle genes. Given that there are so many upstream pathways controlling AKT and also that nc886 interacts with several proteins [1,10], it is challenging to pinpoint a mechanism of how nc886 suppresses AKT. In regard to AKT’s effect on cell cycle genes, FOXO3 might be a link between AKT and CDKN1A, based on our data (Figure 5C) and previous studies (reviewed in [21]). Nonetheless, in the case of CDKN2A and CDKN2C, their regulation by AKT needs another explanation. Neither of them are AKT’s direct target genes (Appendix A). Therefore we sought another possible link of AKT to CDKN2A and CDKN2C. One candidate is the E2F family transcription factors (E2F), because we saw dysregulation of E2F activity in our TFT analysis upon nc886-KD and nc886-EXP (data not shown). AKT has been shown to inhibit the E2F activity [25] and E2F has been shown to activate CDKN2A and CDKN2C [26,27]. Although E2F is generally known to drive G1-S progression when overexpressed, several studies also have reported that E2F TFs not only promote cell cycle progression but also suppress cell cycle progression as a checkpoint or a negative feedback mechanism (reviewed in [28]). Collectively from all these reports, we illustrate sequential events upon nc886 silencing during ESCC carcinogenesis as follows: AKT activation, FOXO and E2F suppression, down-regulation of CDK inhibitors, promotion of the G1-S transition, and eventually fast cell proliferation (Figure 8). We propose the AKT and FOXO3/E2F nexus as a mechanism of how nc886 enhances the expression of CDK inhibitor genes. However, it is challenging to unravel the causal relationship between individual events experimentally by overexpression or KD approaches, because all the players (AKT, FOXO3, and E2F) are multifaceted pathways that will impose pleiotropic effects on cells.

Current staging systems and biomarkers are limited in their capability to assess the risk of a recurrence and benefit from appropriate treatment in ESCC. As cyclin D1-CDK4/6 is crucial for G1 cell cycle progression and generation of neoplastic cells, targeting CDK4/6 could be effective for ESCC patients [29,30,31]. Recently, palbociclib (PD-0332991), the first FDA-approved small molecule CDK4/6 inhibitor, has undergone extensive investigations in pre-clinical studies and clinical trials in a series of human malignancies, including EC [32]. Our mechanistic investigation of nc886, AKT, and cell cycle may represent a tool that could help further refine treatment decisions based on the gene expression profile of a tumor. Using integrated analysis, we identified new pathway-dependent prognostic subgroups of ESCC that showed significant differences in patient survival. This study provides new knowledge by unraveling epigenomic regulation of a signaling pathway by nc886, highlighting nc886-associated genes, and enabling further refinement in sub-classification for improved personalization of treatment for this deadly disease. Furthermore, delivery of nc886 might deserve therapeutic consideration, provided that nc886^−^ ESCC cells appear to be addicted to the nc886^−^ status for their proliferation. In conclusion, this study provides insights into the molecular/genetic mechanism of ESCC oncogenesis and supporting evidence for therapeutics.

## Figures and Tables

**Figure 1 cells-09-00801-f001:**
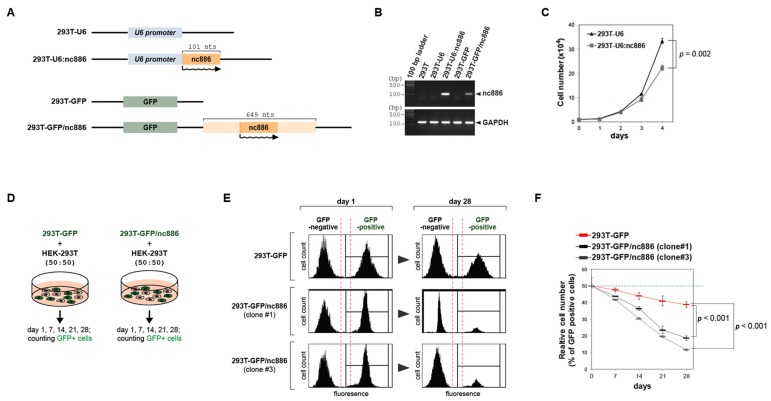
nc886 inhibits cell growth when ectopically expressed in 293T cells. (**A**) Simplified diagrams depicting the feature of plasmids used to construct pairs of an nc886^+^ cell line and the corresponding control nc886^−^ line from the parental 293T cells (see the text for details). Names of cell lines are indicated on the left. nc886 transcripts are illustrated by wavy lines, with an arrowhead designating the transcriptional direction. (**B**) RT-PCR measurement of nc886 and GAPDH as a control. (**C**) Cell proliferation shown by cell numbers. After plating the same number of cells at day 0, cell numbers were counted every indicated day. Mean values and standard errors from triplicate samples are shown, with a *p*-value at day 4 indicated. (**D**) An illustration of the cell mixing experiments for panels E,F. (**E**,**F**). Fluorescence activated cell sorting (FACS) profiles for counting green fluorescent protein (GFP)-negative and -positive cells. Representative FACS data at day 1 and 28 are shown in panel E. At indicated days, % of GFP-positive cells were calculated from the FACS data and are plotted (panel F). Red line designates 293T-GFP; black and grey lines designate clone #1 and #3 of 293T-GFP/nc886. *p*-values (from triplicate samples) are for 28-day samples.

**Figure 2 cells-09-00801-f002:**
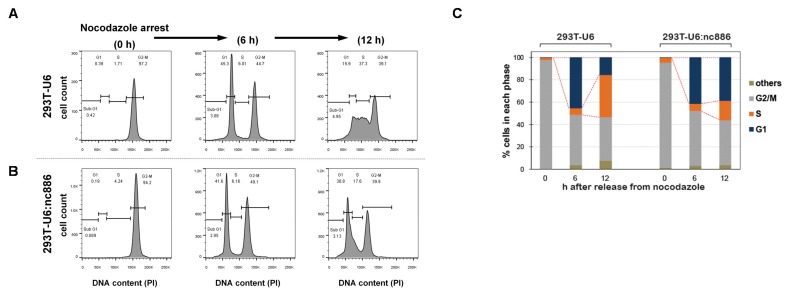
nc886 arrests cell cycle at the G1 phase. (**A**,**B**) FACS cell cycle profiles shown by propidium iodide (PI) staining of DNA contents. Cells were treated with nocodazole and were harvested just before release (0 h) and at indicated time points afterward. From the FACS data (**B**), cells at G1, S, G2, and other (mainly sub-G1) phases were counted and each fraction was calculated (**C**, bar graph). A representative result from duplicate experiments is shown.

**Figure 3 cells-09-00801-f003:**
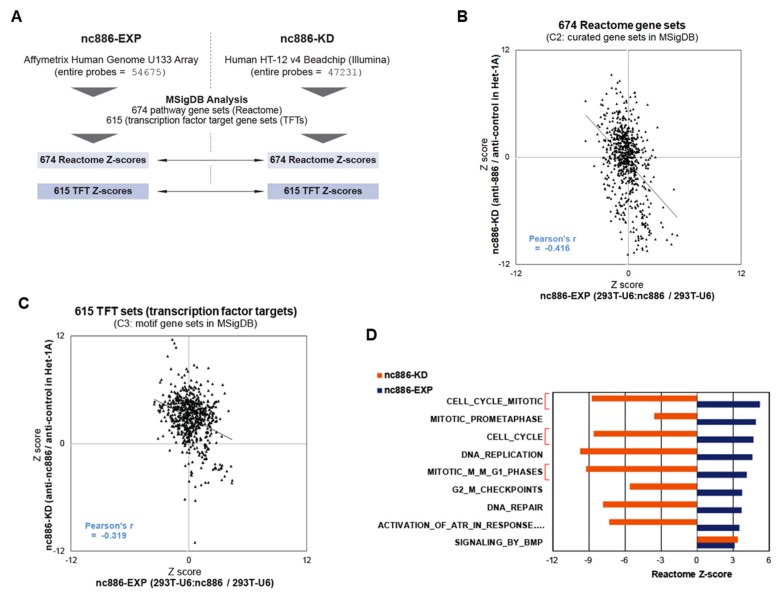
Pathway analyses from the array data of nc886-EXP and nc886-KD. (**A**) Summary of array platforms and analysis workflow. (**B**) A scatter plot of Z-scores for 674 Reactome gene sets that shows the correlation between nc886-EXP (x-axis) and -KD (y-axis). (**C**) A scatter plot of Z-scores for 615 transcription factors (TFs) target gene sets (TFTs), as described in panel B. (**D**) A bar graph showing Z-scores for Reactome gene sets. Nine pathways with a significantly enriched Z-score (>3) were selected from nc886-EXP and were plotted together with Z-scores of nc886-KD.

**Figure 4 cells-09-00801-f004:**
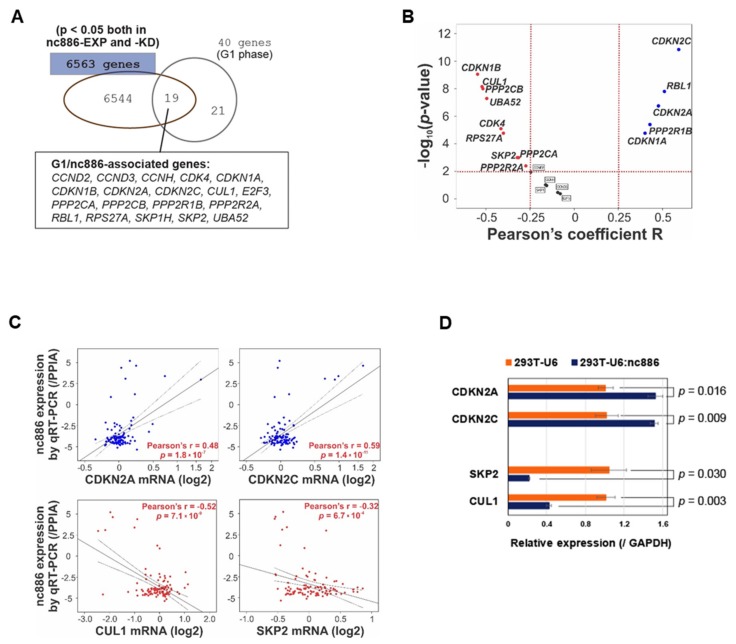
nc886 controls the expression of cell cycle genes. (**A**) Venn diagram of cell cycle genes (40 genes) and significantly altered genes by nc886 (6563 genes). (**B**) Volcano map of 19 nc886/G1-associated genes to show the statistical correlation with nc886 expression in the 108 esophageal squamous cell carcinoma (ESCC) patient cohort. (**C**) Correlation analysis between nc886 expression and representative cell cycle genes. (**D**) qRT-PCR of indicated genes. Mean values from triplicate samples are shown in a bar graph with standard errors and *p*-value indicated.

**Figure 5 cells-09-00801-f005:**
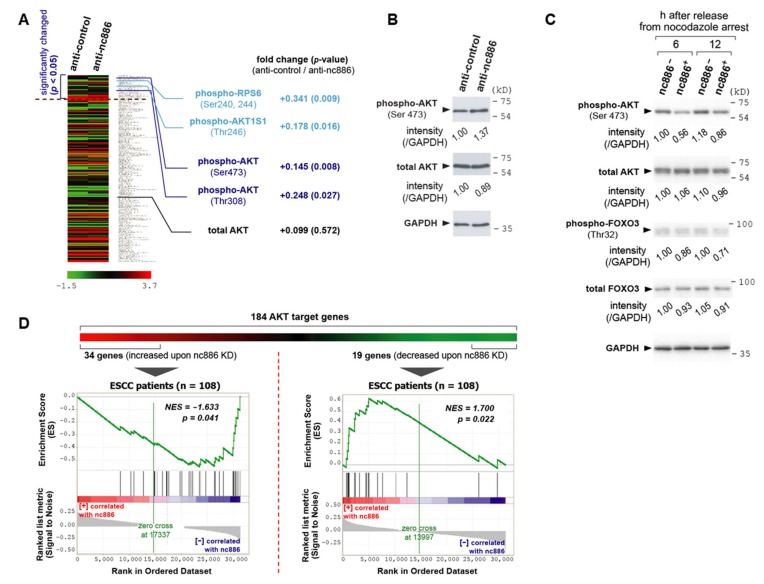
nc886 suppresses the AKT pathway. (**A**) A heat map showing the reverse-phase protein arrays (RPPA) data from nc886 KD of an ESCC cell line, TE1. RPPA values of a total of 172 proteins were sorted by *p*-values in descending order, to locate significantly (*p* < 0.05) changed 23 proteins on the top. Fold change and *p*-values are shown for phospho- and total-AKT as well as AKT target proteins. (**B**) Western blot of phospho- and total-AKT together with GAPDH as a loading control after nc886 KD. Each indicated anti-oligo was transfected into an ESCC cell line, TE-8, and cells were harvested at 48 h after transfection. Each band was quantified with AlphaView software 2.0.1.1 (Alpha Innotech Corp, Santa Clara, CA) and normalized as indicated. (**C**) Western blot as described in panel B. nc886^−^ and nc886^+^ designate 293T-U6 and 293T-U6:nc886 cells respectively. These cells were synchronized by nocodazole, released from G2/M arrest, and harvested at indicated times afterward. (**D**) GSEA to determine whether a priori defined set of AKT genes in nc886 KD shows statistical concordance in ESCC patients according to nc886 expression.

**Figure 6 cells-09-00801-f006:**
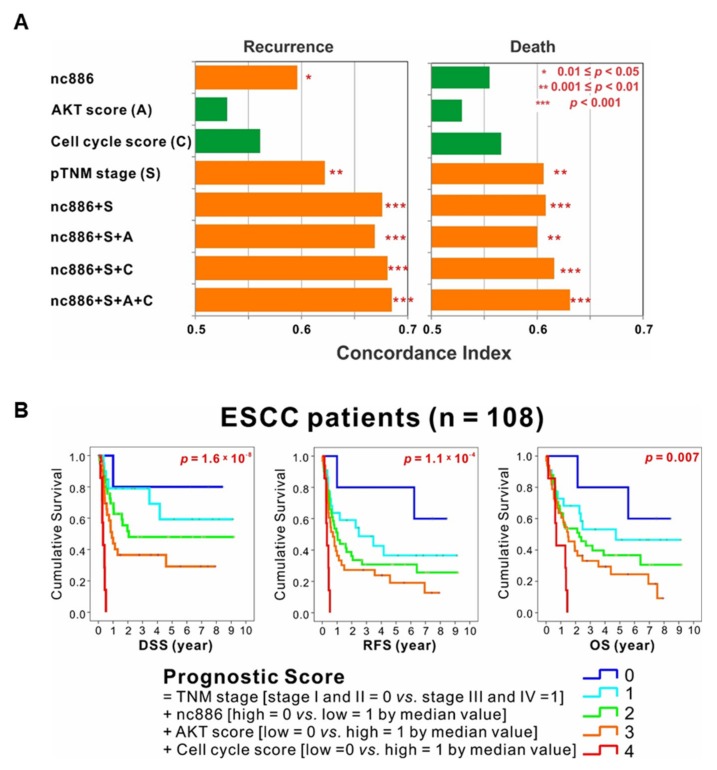
Prognostic value of nc886 and its associated AKT and cell cycle genes. (**A**) Concordance index showing that the combination of nc886, AKT signature score, cell cycle signature score, and tumor-node-metastasis (TNM) staging had the most powerful predictor of recurrence and survival. (**B**) Stratification by prognostic score, which is calculated as the sum of the TNM stage (score 0 for stage I and II vs. score 1 for stage III and IV), the nc886 expression (score 0 vs. 1 respectively for high vs. low expression divided by the median value), the AKT signature score (score 0 vs. 1 for low vs. high by the median value), and the cell cycle score (score 0 vs. 1 for low vs. high by the median value), showing distinct disease-specific survival (DSS), recurrence-free survival (RFS), and overall survival (OS) curves in ESCC patient cohort.

**Figure 7 cells-09-00801-f007:**
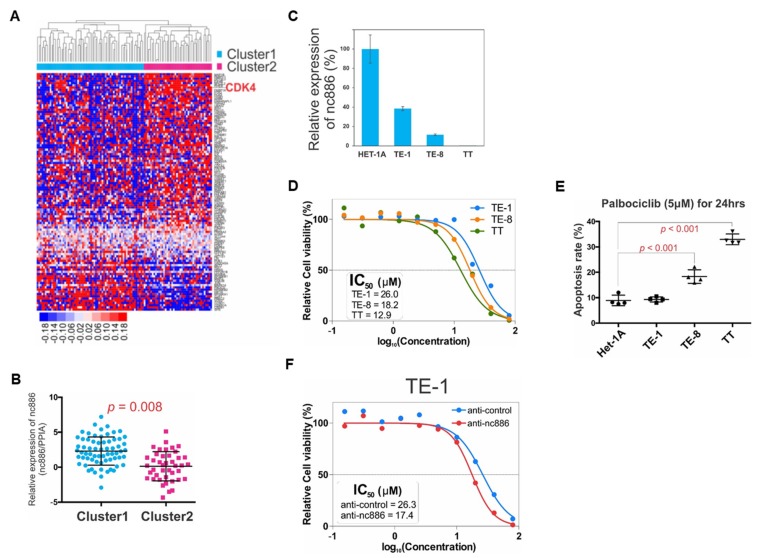
Prognostic and predictive values of nc886. (**A**) Unsupervised clustering of AKT and cell cycle genes in ESCC cohort (*n* = 108). There are two distinct clusters. Majority genes including CDK4 are up-regulated in Cluster2. (**B**) nc886 expression between Cluster1 and Cluster2. nc886 is significantly lower in Cluster2. (**C**) nc886 expression in esophageal cell lines by qRT-PCR. The nc886 expression of indicated cell lines, which was calculated by 2^−ΔΔCt^ using cyclophilin A as a normalization control, is shown relative to Het-1A. (**D**) MTT assay after palbociclib treatment in three ESCC cell lines. % of cell viability, relative to no palbociclib treatment, is plotted against the drug concentration to calculate IC_50_ values. (**E**) Apoptosis assay of esophageal cell lines showing an inverse correlation of nc886 expression levels to the sensitivity to palbociclib. (**F**) Palbociclib treatment after nc886 KD. Cells were transfected with the indicated anti-oligo. At 48 h after transfection, palbociclib was treated for 24 h and then MTT assays were performed as described in Materials and Methods.

**Figure 8 cells-09-00801-f008:**
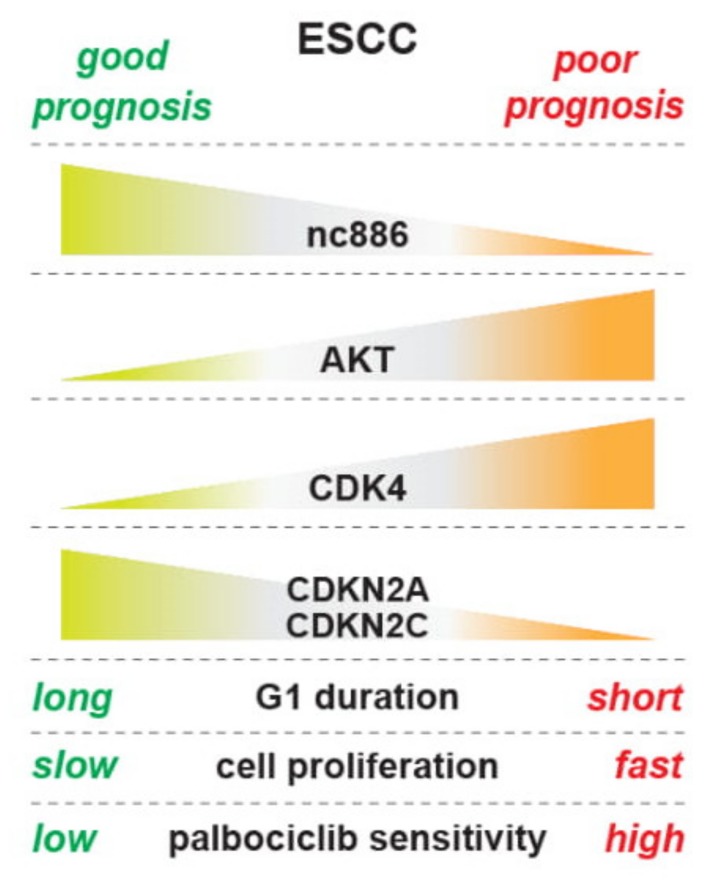
An illustration summarizing results in this study.

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
