# Peer review of "A Regulatory Noncoding RNA, nc886, Suppresses Esophageal Cancer by Inhibiting the AKT Pathway and Cell Cycle Progression"

_cells, 2020, doi:10.3390/cells9040801_

Round 1

Reviewer 1 Report

The authors adequately addressed my concerns.

Reviewer 2 Report

I reviewed with interest the revised version of the manuscript “cells-689847” by Wonkyun Ronny Im and colleagues about the functional characterization of non-coding RNA nc886 in esophageal squamous cell carcinoma (ESCC).

As stated earlier, the study of nc886 is of particular interest given that its function seems to be cancer dependent. Overall the study is well designed and the experimental data support the conclusions made by the authors. The method section is clearly written. The statistical analysis is clear as well. The clinical relevance of the study is also established in a large cohort of ESCC patients.

Importantly, most of the issues raised by the reviewers have been addressed. In particular, additional experiments have been performed to clarify specific points and to strengthen the data (newly added data are Fig 5B, 7D, 7F, S1, S3 and 5C; improved figures are Fig 1F and 5C).

This manuscript is a resubmission of an earlier submission. The following is a list of the peer review reports and author responses from that submission.

Round 1

Reviewer 1 Report

I reviewed with interest the manuscript “cells-689847” by Im et al about the functional characterization of non-coding RNA nc886 in esophageal squamous cell carcinoma (ESCC). The authors previously reported that nc886 silencing in ESCC patients is associated with a shorter survival. Here, by gain and loss of function experiments in vitro and transcriptomic analysis, the authors show that nc886 delays G1-S transition by controlling cell cycle genes (e.g. CDKN2A, CDKN2C, SKP2, CUL1). They further report that nc886 targets AKT. Thus, nc886 silenced cells exhibit active AKT, higher proliferation and are more sensitive to palbociblib, a drug targeting CDK4/6. The authors also report a set of nc886-associated genes predictive of patient survival in ESCC.

The study of nc886 is of particular interest given that its function seems to be cancer dependent. Overall the study is well designed and the experimental data support the conclusions made by the authors. The method section is clearly written. The statistical analysis is clear as well. The clinical relevance of the study is also established in a large cohort of ESCC patients. However, several points should be addressed to strengthen the study.

Comments

It is not clear why two microarray platforms were used in the study. It seems that part of the data was previously published. This point needs to be clarified in the method section. The dose of 5µM palbociclib should be justified. Figure 1F: please include statistical analysis. Similarly, the number of replicates should be indicated in Figure 2. Section 3.3 is rather difficult to read. Gene expression signatures obtained after nc886 gain and loss of function should be provided and directly compared (i.e. Venn diagram and illustrated by a heatmap) even though the kinetics are not similar. This comparison should be informative. Section 3.4, line 4: the term “arrayed” is somehow confusing. The molecular mechanisms linking nc886 and altered expression of AKT and cell cycle genes should be discussed (ceRNA sponging miRNA targeting AKT?) In Figure 7 the authors show an inverse correlation between nc886 expression and sensitivity to palbociclib. However, sensitivity to the drug should be determined after gain and loss of function of nc886 to strengthen the data.

Reviewer 2 Report

My major concern regarding this manuscript is that most of the experiments were performed using only HEK293T cells. While nothing is wrong with using such a cell lines, in order to claim that a ncRNA is functional in cancer one is expected to demonstrate such a function in (at least) two tumor cell lines.

I am aware that some of the experiments were performed using esophageal cancer cell lines however, as is the case in Fig 5 for example, only one esophageal cancer cell lines was used.